# Coastal Adaptation to Climate Change and Sea-Level Rise

**Gary Griggs [1],\*** and **Borja G. Reguero [2]**

[1] Department of Earth Sciences, University of California Santa Cruz, Santa Cruz, CA 95064, USA
[2] Institute of Marine Sciences, University of California Santa Cruz, Santa Cruz, CA 95064, USA; breguero@ucsc.edu
\* Correspondence: griggs@ucsc.edu

**Abstract:** The Earth's climate is changing; ice sheets and glaciers are melting and coastal hazards and sea level are rising in response. With a total population of over 300 million people situated on coasts, including 20 of the planet's 33 megacities (over 10 million people), low-lying coastal areas represent one of the most vulnerable areas to the impacts of climate change. Many of the largest cities along the Atlantic coast of the U.S. are already experiencing frequent high tide flooding, and these events will increase in frequency, depth, duration and extent as sea levels continue to rise at an accelerating rate throughout the 21st century and beyond. Cities in southeast Asia and islands in the Indo-Pacific and Caribbean are also suffering the effects of extreme weather events combined with other factors that increase coastal risk. While short-term extreme events such as hurricanes, El Niños and severe storms come and go and will be more damaging in the short term, sea-level rise is a long-term permanent change of state. However, the effects of sea-level rise are compounded with other hazards, such as increased wave action or a loss of ecosystems. As sea-level rise could lead to the displacement of hundreds of millions of people, this may be one of the greatest challenges that human civilization has ever faced, with associated inundation of major cities, loss of coastal infrastructure, increased saltwater intrusion and damage to coastal aquifers among many other global impacts, as well as geopolitical and legal implications. While there are several short-term responses or adaptation options, we need to begin to think longer term for both public infrastructure and private development. This article provides an overview of the status on adaptation to climate change in coastal zones.

**Keywords:** sea-level rise; climate change; shoreline erosion; adaptation; managed retreat

## 1. Introduction

The climate has been changing for as long as we have had the Earth and the Sun. The amount of solar radiation we receive from the Sun has varied over tens of thousands of years due in large part to the Milankovitch cycles, which control the distance between the Earth and the Sun. Sea level is intimately tied to ocean warming and therefore climate change. As temperatures rise during warm or interglacial periods, seawater expands, and the ice covering Antarctica, Greenland and the mountain glaciers of the planet melts, increasing sea levels globally. During cooler (glacial) periods, sea level is lowered as seawater cools and takes up less volume, and more precipitation falls as snow freezes to ice and allows ice sheets and glaciers to expand.

However, global warming from greenhouse gas concentration during the last century has led to higher sea levels, globally driven by the melting of ice sheets and the thermal expansion of the ocean. Sea levels will continue to rise in the future, critically threatening low-lying coastal zones [1]. The potential for high-end sea-level rise may remain despite the ambition of the Paris Agreement to limit global temperature increase well below 2 °C above pre-industrial levels, given the inertia in ocean processes.

Coastal zones are particularly vulnerable to the impacts of sea-level rise. However, sea-level rise is not the only way climate change affects coastlines. Climate changes also affect

processes and dynamics in coastal zones through interannual and long-term changes in winds, storm surges or wave action [2–6]. Changes in wind patterns, wave power, extreme waves and sea levels [7–12] all drive important effects on coastlines. However, these changes vary between regions and coastlines, from seasonal to interannual and long-term temporal scales, triggering different impacts locally, such as flooding and erosion [3,13,14]. Expected rises in water temperatures and ocean acidification will also impact coastal ecosystems, with important implications for the services they provide, such as fisheries, coastal protection or carbon sequestration [15–22].

Increasing coastal hazards combined with development and demographic concentration in coastal areas makes the need for adaptation urgent. However, the status of implementation is still limited along many coastlines, challenged by important technical, economic, financial and social factors [23]. Coastal communities require targeted responses, plans and informed action in order to address the present and future effects and costs from sea-level rise and climate change.

This paper is intended to serve as a comprehensive introduction or overview to this Special Issue of WATER on Coastal Adaptation to Climate Change and Sea-Level Rise. Here, we provide a summary of the current issues and challenges involved in this important and timely subject. The article describes existing knowledge on coastal hazards, their effects in coastal areas and adaptation responses. The article also describes the main challenges and current advances. We rely on recent research and examples from many regions. The article aims to provide an overview on coastal adaptation that can provide the reader with a useful document for future reference.

## 2. Sea-Level Rise

### 2.1. Historic Changes in Sea Level

Geological evidence, primarily from sediments and fossils collected from the continental shelf, provides clear global confirmation that the rapid rise of the sea level following the end of the last ice age approximately 20,000 years ago slowed to nearly a halt approximately 7–8000 years ago [24]. From that time until around the mid-1800s, sea level rose at less than 1 mm/year. With the onset of the industrial era and the increasing combustion of fossil fuels (coal, oil and natural gas), the greenhouse gas content of the atmosphere gradually increased. Since the onset of the industrial revolution the content of carbon dioxide in the atmosphere has increased from natural levels varying from approximately 175–275 ppm to 419 ppm today, which is an increase of approximately 50 percent. Greater greenhouse gas concentrations have amplified the Earth's natural greenhouse effect, leading to a gradually warming planet. Over the past 100 years, the Earth's climate has warmed by approximately 1 °C (1.8 °F; Figure 1). As temperatures rose, ice sheets and continental glaciers melted at an increasing rate and seawater warmed and expanded. Global sea levels rose in response, raising sea levels at a more rapid rate than over the previous 7000–8000 years [25].

Today, hundreds of tidal gauges around the coastlines of the world are recording sea water levels, but the first measurements date from the mid-1800s. Tide gauges track the local or relative sea level, which is the elevation of the local sea level relative to land motion, including uplift and subsidence. Globally averaging historic records documented sea-level rise values ranging from ~1.2 to ~1.7 mm/year (4.7 to 6.8 in century) over much of the 20st century [26,27]. These tide gauges are not evenly distributed, however, with most in the northern hemisphere (U.S. and Europe). While tide gauges provide relative or local sea-level rise rates, a recent evaluation of 32 tide gauge records from all U.S. coastlines revealed that, with the exception of the U.S. northeast coast and Alaska, every coastal location in the continental U.S. has experienced an upturn in relative sea-level rise rate since 2013–2014, despite wide differences in the magnitude and trending direction of relative sea-level rise acceleration [28].

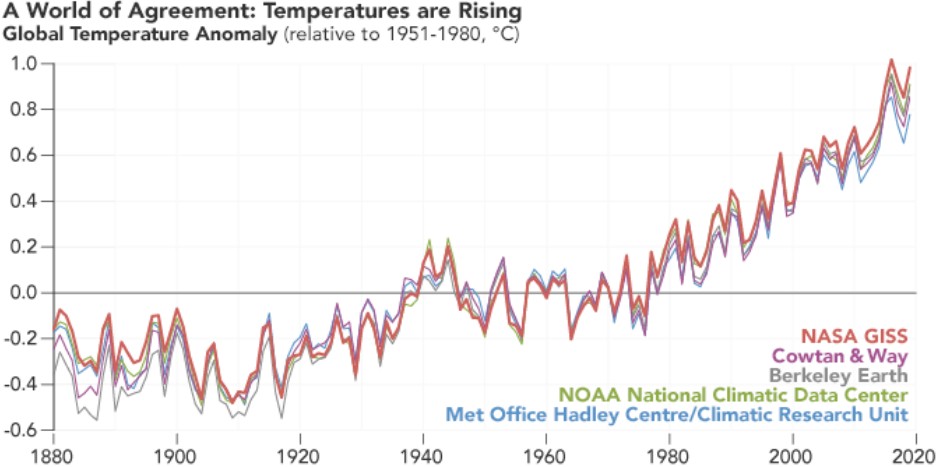

**Figure 1.** Annual global surface temperature 1850–2019. Source: (NASA Earth Observatory/Robert Simmon). The line plot below shows yearly temperature anomalies from 1880 to 2019 as recorded by NASA, NOAA, the Berkeley Earth research group, the Met Office Hadley Centre (United Kingdom), and the Cowtan and Way analysis. NASA's temperature analyses incorporate surface temperature measurements from more than 20,000 weather stations, ship- and buoy-based observations of sea surface temperatures, and temperature measurements from Antarctic research stations. Credits: NASA's Earth Observatory, obtained from: https://earthobservatory.nasa.gov/world-of-change/global-temperatures, accessed on 1 August 2021.

In 1993, two satellites were placed in orbit (Topex and Poseidon), followed by Jason-1, -2 and -3, with the objective of measuring global or absolute sea level accurately and precisely from space using lasers. The average sea-level rise rate from these satellite measurements over their 27 years of operation is now 3.4 mm/year (13.4 in./century), but this rate is accelerating [29,30]. More recently, independent data from European satellites [31] has been used in order to increase both the time period covered (1991–2019) as well as the geographic distribution of data (from 66 degrees to 82 degrees latitude). Satellite-based observations now allow us to measure that the average acceleration of sea-level rise, which has been 0.1 mm/year$^2$ between 1991 and 2019. The average rate of rise of 3.4 mm/year over the past 27 years has now increased to about 4.8 mm/year, or approximately 18.9 in./century (Figure 2), based on observations of the past 10 years (https://www.aviso.altimetry.fr/en/data/products/ocean-indicators-products/mean-sea-level.html) (accessed on 1 July 2021).

*2.2. Future Sea Levels*

Tide gauges and satellite-based observations provide a good understanding of past and present sea level. However, the challenge for coastal regions around the planet is projecting sea-level rise and its impacts into the future. This is an important objective of the Intergovernmental Panel on Climate Change (IPCC), but individual geographic entities (local to national governments) are simultaneously involved in developing future sea-level rise projections for their own regions [32]. Future climate projections are developed through global climate models, which include uncertainties and assumptions of future greenhouse gas emissions (i.e., Representative Concentration Pathways) and model the inputs or factors that will affect global climate, including ice melt and consequently sea-level rise [25]. Today, the predictions or projections for the next few decades are in general agreement but estimates for the end-of-century vary between models and depend on Representative Concentration Pathways (RCPs), with increasingly wider uncertainties and ranges by 2100. The latest estimates indicate that values for the end-of-century (2100) range from a low of ~50 cm (~20 inches) to as high as ~310 cm (~10 feet), as a function of greenhouse gas emission scenarios and various probabilities or uncertainties, especially concerning the extent of Greenland and Antarctica ice melt [33] (Figure 3).

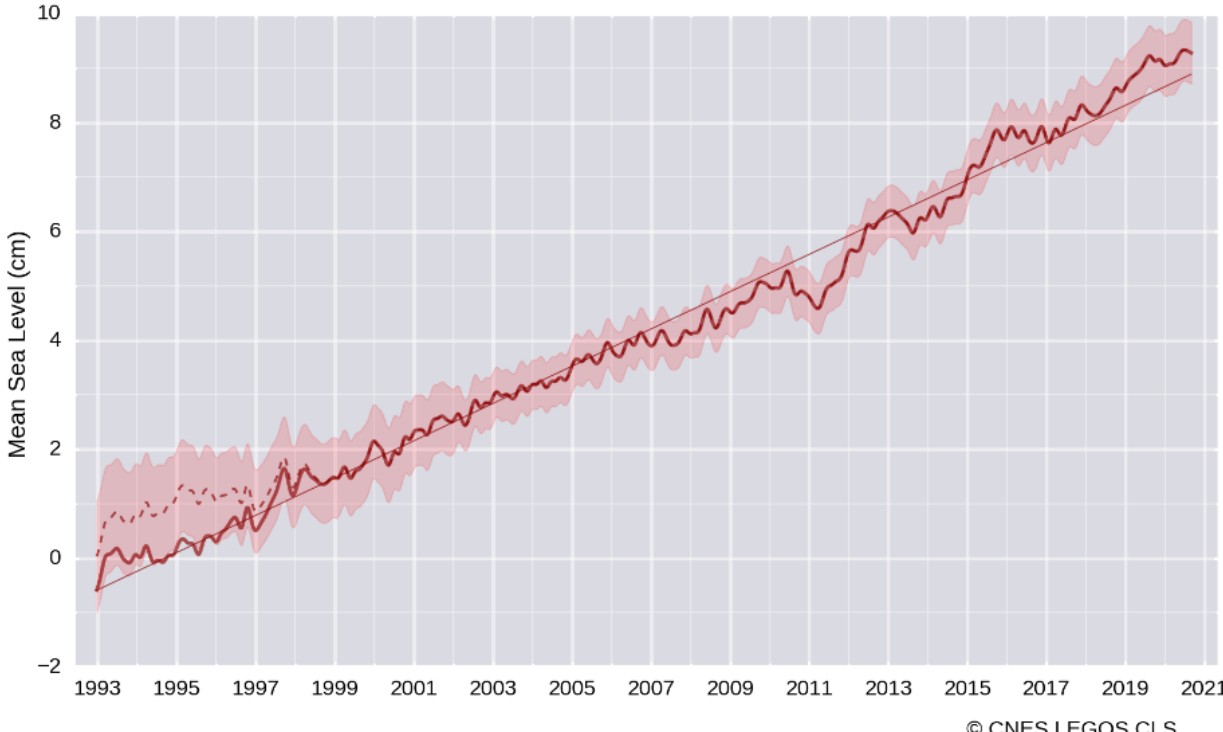

**Figure 2.** Sea-level rise from satellite altimetry 1993–2020. Source: based on [30], obtained from: https://www.aviso.altimetry.fr/en/data/products/ocean-indicators-products/mean-sea-level.html, accessed on 1 August 2021. The reference global mean sea level (GMSL) is based on data from the TOPEX/Poseidon, Jason-1, Jason-2 and Jason-3 missions from January 1993 to present, after removing the annual and semi-annual signals and applying a 6-month filter. By applying the postglacial rebound correction (−0.3 mm/year), the rise in mean sea level has thus been estimated as 3.4 mm/year (straight line on the figure).

Understandably, while projections of future sea levels typically only extend out to 2100 due to increasing uncertainties, sea-level rise will not stop then, but will likely continue for decades and even centuries into the future. Even in the absence of further greenhouse emissions, the sea-level rise inertia will continue, and sea levels will increase in the future. There is approximately 66 m (~216 feet) of potential sea-level rise contained in the ice sheets and glaciers of Antarctica, Greenland and the mountain glaciers of the planet (http://www.antarcticglaciers.org/glaciers-and-climate/estimating-glacier-contribution-to-sea-level-rise/, accessed on 1 August 2021). No one believes that these will all melt this century, but this is the total potential that exists if it were all to melt.

In just this century, raising the sea level just 1 m will create substantial issues for developed shorelines around the planet. A recent global assessment determined that approximately 110 million people live below the present high tide today, and 250 million occupy land below current annual flood levels [34]. For the first few meters of sea-level rise, more than 3 million more people are at risk with each vertical 2.5 cm (one inch) of rise. One billion people today, approximately 13% of the entire global population, live less than 10 m (33 feet) above today's high tide.

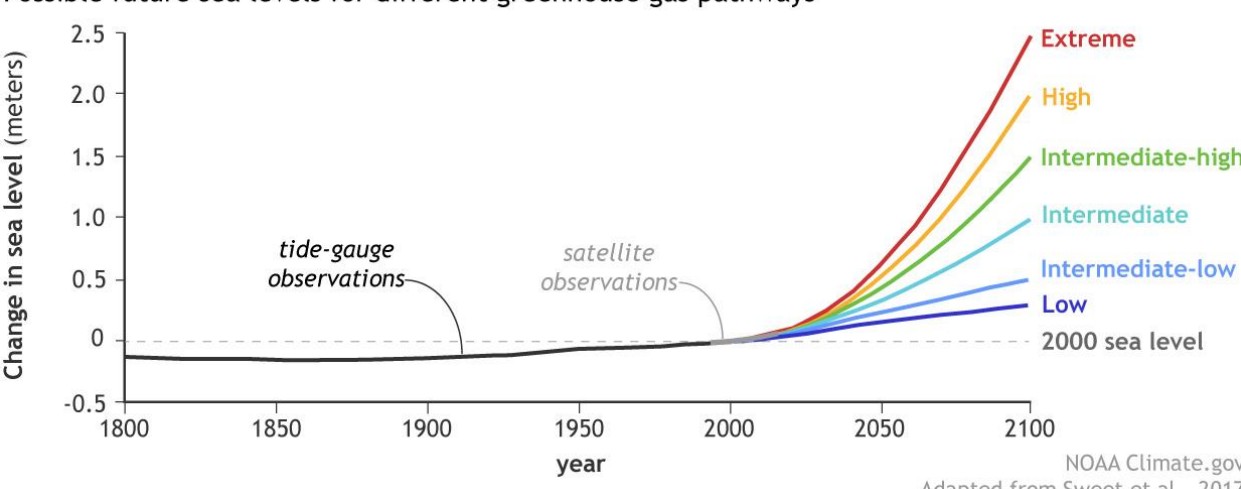

**Figure 3.** Possible future sea levels for different greenhouse gas emission pathways. Observed sea level from tide gauges (dark gray) and satellites (light gray) from 1800–2015 show historical trajectory. The scenarios differ based on potential future rates of greenhouse gas emissions and differences in the plausible rates of glacier and ice sheet loss. Source: NOAA Climate.gov graph, adapted from Figure 8 in [35].

## 3. Effects of Climate Change and Sea-Level Rise in Coastal Areas

### 3.1. How Future Sea-Level Rise Will Affect Coastal Areas

Looking to the future, the potential loss of public infrastructure and private development due to sea-level rise will have enormous economic impacts on coastal nations globally [1]. Different coastal environments face unique hazards, however, as a result of their geology and topography, regional climatic settings and development patterns. Coasts display a variety of landforms (e.g., estuaries, beaches, dunes, low bluffs, high cliffs and steep mountains) and also differing development patterns (low to high density). Lower-lying shoreline areas are more vulnerable to flooding from wave action, hurricanes and large storm waves acting simultaneously with very high tides and atop the higher sea levels of the future. Higher-elevation areas, such as bluffs, cliffs and coastal mountains, are more vulnerable to coastal erosion from wave attack during high tides or elevated sea levels. Nonetheless, higher sea levels in the future will mean: (1) more frequent and higher elevation flooding of low relief shoreline areas [36,37], followed by permanent inundation and loss of beaches and coastal wetlands [14,38]; and (2) waves reaching and impacting the base of coastal cliffs, bluffs and dunes more often, leading to increased erosion rates.

The economic impacts of coastal hazards will also vary with the degree and type of development and whether it is public or private. Passive erosion, or the gradual loss of beaches from continuing sea-level rise where the back beach has been fixed by a seawall, rock revetment or some other structure, will be a major challenge along highly developed and armored coasts [24,38,39]. Along the intensively developed ~325 km (233-mile) coastline of southern California, for example, where millions of people use the beaches, 38 percent of the entire shoreline has now been armored (Figure 4), and with rising sea levels, the issue of passive erosion and beach loss will become more pressing [40].

### 3.2. Nuisance Flooding

To date, much of the research on the impacts of sea-level rise has focused on the occurrence and damage of sea level extremes, such as from tropical cyclones or other storms [41–43], as sea-level rise contributes to more flooding by increasing the probability of extreme floods [36,44]. However, nuisance flooding (also known as sunny day floods) has increased on U.S. east coasts in recent decades due to sea-level rise [45,46].

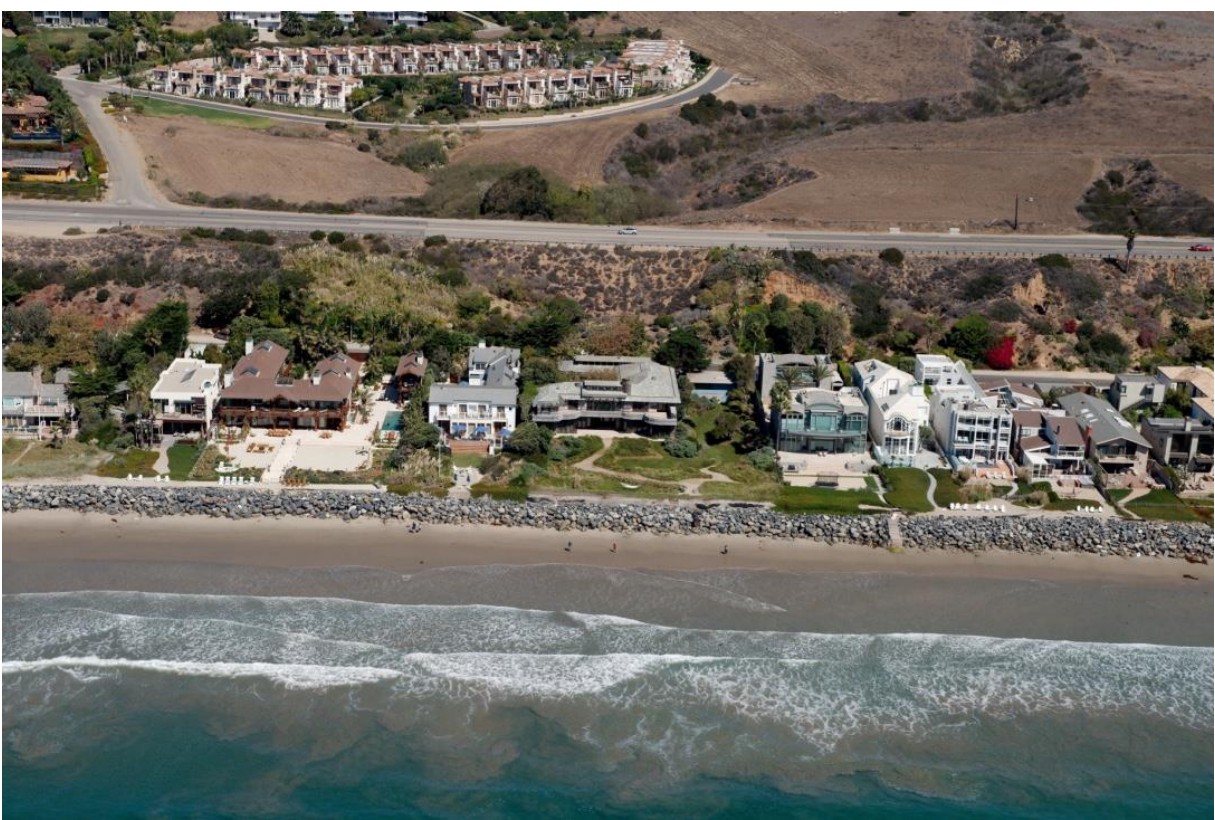

**Figure 4.** Rip-rap revetment armoring a section of the Malibu, California (USA) shoreline. Courtesy: California Coastal Records Project.

Coastal nuisance flooding is considered to be minor flooding from the sea that causes problems such as flooded roads and overloaded stormwater systems, which can be major inconveniences for people and provide a habitat for bacteria and mosquitoes. Based on over 70 years of observations from the U.S., a study found that the total number of nuisance flooding events caused by tidal changes have increased at an exponential rate since 1950, adding 27% more nuisance flooding occurrences in 2019 [47]. Estuaries show the largest changes because of tide changes associated with anthropogenic alterations, such as the dredging of channels, land reclamation, changes in river flows and other developments.

Frequent high-tide flooding also affects local economic activity. For example, a study in Annapolis, Maryland, found that frequent high-tide floods have reduced visits to the historic downtown by 1.7%, but with 8 and 30 cm (3 and 12 inches) of additional sea-level rise, high-tide floods would reduce visits by 3.6% and 24%, respectively [48]. The impacts of high-tide flooding should also be better characterized and understood in order to help guide efficient local responses and include them in urban planning.

## 4. Short-Term Coastal Hazards versus Long-Term Sea-Level Rise

Climate change will also influence coastal hazards besides sea-level rise. While considerable research and planning effort today is focused on increasing the accuracy of future sea-level rise projections, in the short- or near-term (until perhaps the mid-century), it will likely be the extreme events that will be more damaging to coastal development and infrastructure [40]. These events include cyclones, typhoons and hurricanes, large storm waves arriving simultaneously with very high tides or elevated water levels and tsunamis.

### 4.1. Interannual Changes

The large El Niño of 1982–83 raised sea levels along the California (USA) coast, as recorded at tide gauges, to the highest values ever registered, ranging from 29 cm (11.4 in.) above predicted tidal heights at San Diego, California, 32.3 cm (12.7 in.) at Los Angeles and

53.9 cm (21.2 in.) at San Francisco. These extreme El Niño-related tides were the highest water levels recorded during the prior 77 years in San Diego, 59 years in Los Angeles and 128 years in San Francisco. Using average rates of global sea-level rise from satellite altimetry (3.4 mm/year or 13.4 inches/century), these elevated 1983 El Niño water levels were equivalent to 85, 95 and 158 years of sea-level rise at recent rates, respectively, at these three locations. In addition to elevated water levels, seven major storms brought large waves during periods of high tides. These combined to produce USD 265 million in coastal damage to oceanfront property (in 2020 USD). Damage was not restricted to just broken windows and flooding of low-lying areas—33 oceanfront homes were totally destroyed and dozens of businesses, parks, roads and other public infrastructure were heavily damaged (Figure 5).

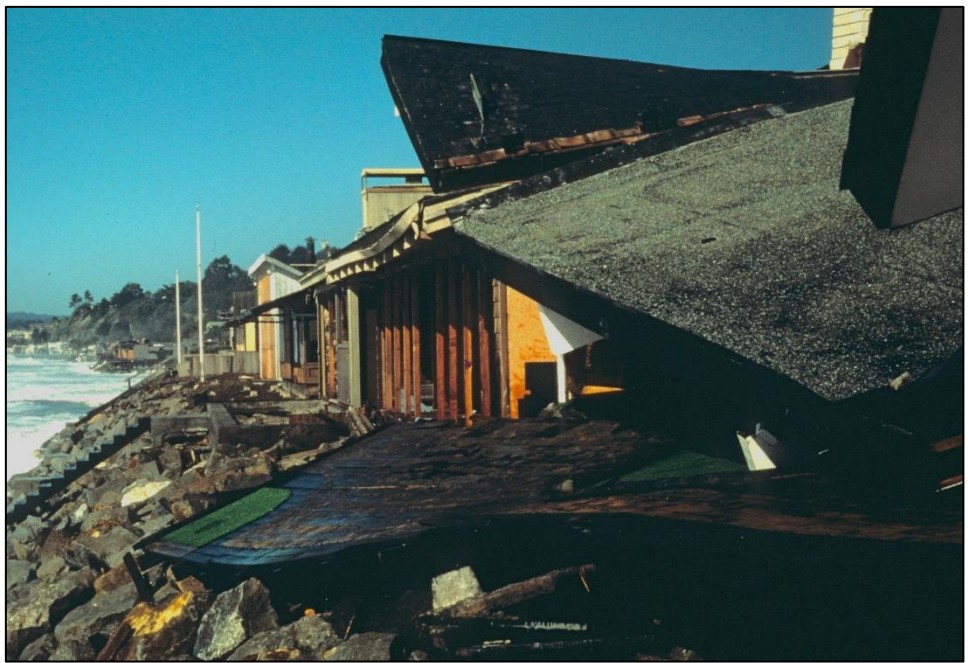

**Figure 5.** Storm waves arriving simultaneously with high tides broke through the front of these homes built along the northern Monterey Bay, California, shoreline (Photo: Gary Griggs).

Yet again, in mid-January 1988, very large waves struck the southern California coast suddenly and left USD 62 million in losses [40]. A decade later, during the El Niño winter of 1997–98, intense rainstorms hit southern California, washing out roads and railroad tracks, overflowing flood control channels and battering the coast, leading to 17 fatalities and over half a billion dollars ($10^9$) in damage [49]. Recent El Niños along the US West Coast and Pacific have also set new records and caused widespread flooding and erosion [3,4,50].

*4.2. Hurricanes, Cyclones and Typhoons*

While other coastal areas around the planet do not experience the impacts of El Niños, they have their own extreme events to contend with. On 8 November 2013, Typhoon Haiyan, the strongest tropical cyclone ever to make landfall based on wind velocities, cut a devastating swath across the central Philippines in the tropical western Pacific. The storm strength was equivalent to a Category 5 hurricane (the highest level of intensity) with sustained wind speeds at landfall of 312 km/h. (195 mph), the highest ever recorded, and gusts of up to 376 km/h. (235 mph). When wind speed reaches approximately 192 km/h. (120 mph), it is no longer possible for a human being to stand up. Thirteen percent of the nation's entire population, nearly 13 million people, was affected. There were at least 6300 fatalities and 28,700 injuries. As a result of the lightweight construction materials

commonly used in the Philippines and the extreme wind velocities, over 281,000 houses were reported as destroyed, with 1.9 million people displaced (Figure 6).

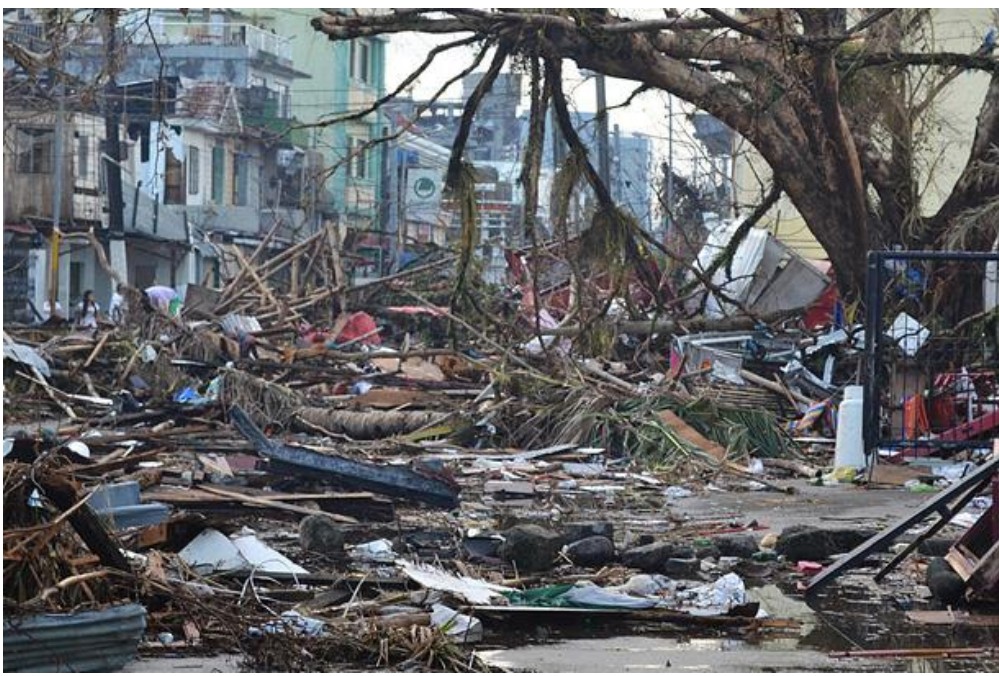

**Figure 6.** Damage from Typhoon Haiyan in the Philippines, 2013 (Photo: Trocare via Wikimedia).

A year earlier, Superstorm Sandy produced a record storm surge in New York City. The water level at the southern tip of Manhattan (New York, NY, USA) topped 4.2 m (13.9 feet), exceeding the 3.1 m (10.2 feet) record set by Hurricane Donna fifty-two years earlier, driven by the size and angle of superstorm Sandy along the U.S. East Coast. For perspective, the average long-term sea-level rise rate for the NOAA tide gauge at the southern end of Manhattan Island has been 2.87 mm/year. since 1856. Seventy-two lives were lost and losses reached USD 50 billion from damage to homes and other buildings, roads, boardwalks and mass transit facilities in low-lying coastal areas of both New York and New Jersey (USA) from storm surges and large waves.

Tropical cyclone-induced coastal flooding will worsen under climate change from the combined effects of sea-level rise and changes in storm activity. For the U.S., the compound effects of SLR and tropical cyclone climatology changes will turn the historical 100-year flood levels into annual events in the New England and mid-Atlantic regions and 1–30-year events in the southeast Atlantic and Gulf of Mexico regions in the late 21st century [51]. Even in regions where the effect of strengthened storms could be compensated with the displacement of storm tracks, as in New York, the effects of higher mean sea levels will drive significant increases in flood levels [52], with important consequences for coastal risk and adaptation needs.

Apart from geographic coastline variability and sea-level rise, whether climate change will drive a future increase in the frequency and magnitude of climatic patterns, such as El Niños or hurricanes events, remains uncertain. There seems to be an emerging consensus, however, that warmer surface ocean water will raise evaporation rates and increase the frequency and magnitude of hurricanes, cyclones and typhoons, potentially delivering more damage when they make landfall. NOAA has suggested that an increase in Category 4 and 5 hurricanes is likely (https://www.gfdl.noaa.gov/global-warming-and-hurricanes/, accessed on 1 August 2021), with hurricane wind speeds increasing by up to 10%, and with 10–15% more precipitation in a 2 °C scenario.

Recent storms, such as Hurricane Harvey (2017), which dropped over 152 cm (60 inches) in some locations near Houston, Texas (USA), Florence (2018), with over 89 cm (35 inches),

and Imelda (2019; 112 cm/44 inches), in addition to the impacts of wind, storm surges and wave action, as demonstrated with Hurricanes Irma and Maria in the Caribbean in 2017, demonstrate the devastating effects that can be triggered by more frequent, intense or wetter hurricanes. The connection between climate change and hurricane frequency is less straightforward. It is likely that the number of storms will remain the same or even decrease, with the primary increase being in the most extreme storms. Areas affected by hurricanes are also shifting poleward, likely with expanding tropics due to higher global average temperatures. The changing patterns of tropical storms (a shift northward in the Atlantic) could also put more property and human lives at risk, but much more research is required to better characterize and predict how such patterns will change in the future. Independently, historic data show how the cost of hurricanes is also increasing globally, associated with a conjunction of climate, more intense coastal development and other factors [53].

### 4.3. Wave Action

Changes in wave action can be one of the most important drivers of coastal change. Wave energy has increased and is globally associated with global warming [7]. Increases in wave heights are more significant for higher percentiles, especially in high latitudes [11]. Future projections also point to increases in wave heights along many coastlines, including the extremes, although with strong spatial variability [10,12,54]. Local changes in wave action, as well as the strong influence of interannual patterns, such as El Niño or the North Atlantic Oscillation [55–57], necessitate including local projections and effects when predicting and forecasting wave-driven impacts in coastal areas. Furthermore, recent research has also indicated that wave forces could drive important impacts, especially for low-lying areas, sandy shorelines and islands, or coastal infrastructure [58,59]. Changes in wave action will accelerate coastal erosion (Figures 7–9), require upgrading and increased costs of coastal defenses [60,61], port activity and operations [62] and alter sediment transport and beaches [14,63,64]. Local changes in wave action, as well as the strong influence of interannual patterns, such as El Niño or the North Atlantic Oscillation [55,56], need local projections for predicting these local impacts. These impacts are occurring globally in developed regions such as the U.S., but also represent important challenges in developing countries and island nations (Figures 7 and 8).

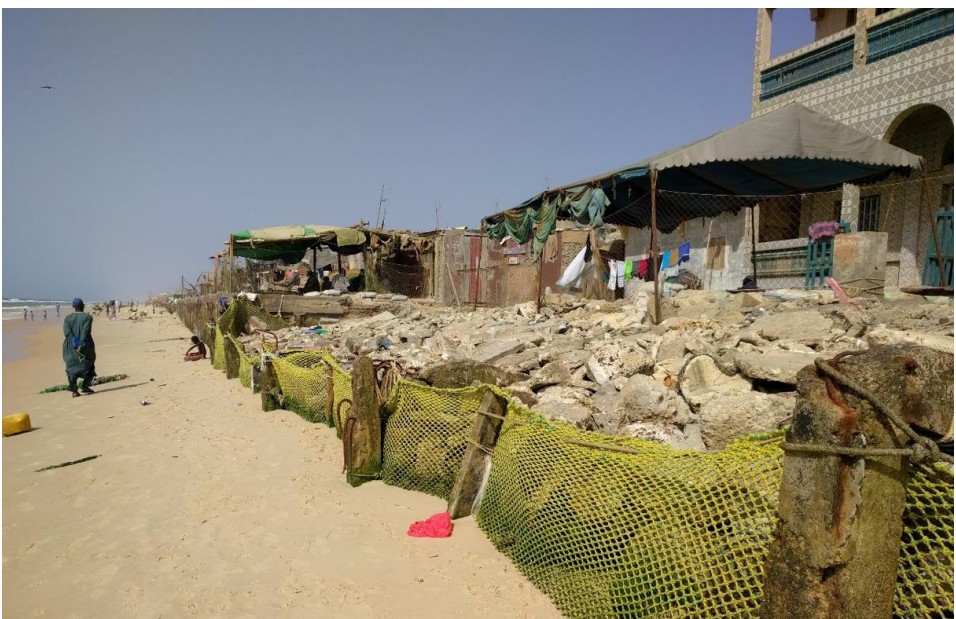

**Figure 7.** Beach erosion in Saint Louis, Senegal, threatens houses and the livelihood of many. Source: B. G. Reguero.

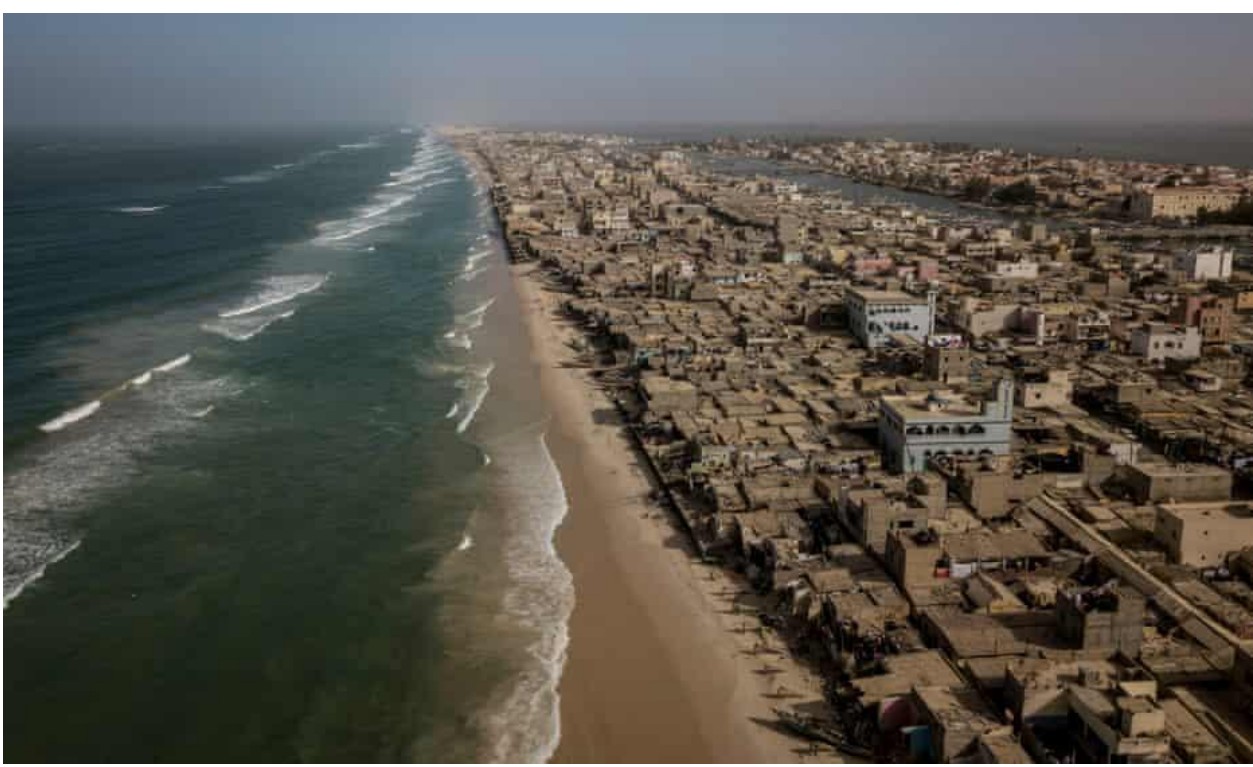

**Figure 8.** Saint Louis, the old colonial capital of Senegal, faces a flooding threat that has already seen entire villages lost to the Atlantic. Source: The Guardian—https://www.theguardian.com/environment/2020/jan/28/how-the-venice-of-africa-is-losing-its-battle-against-the-rising-ocean, accessed on 1 August 2021.

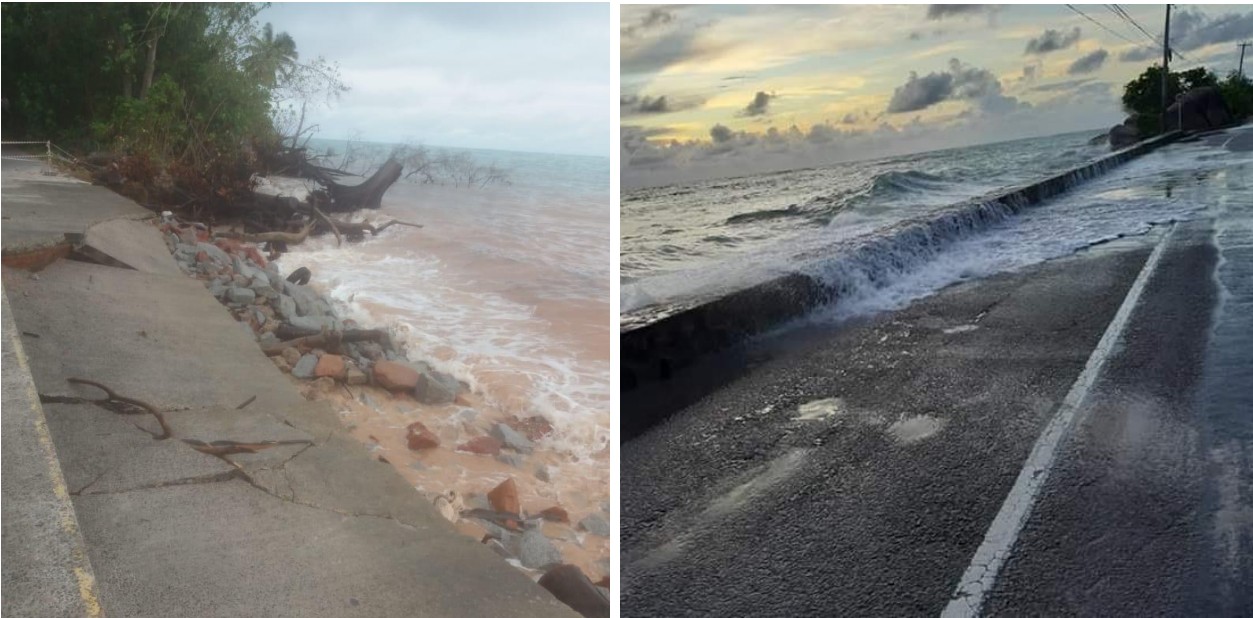

**Figure 9.** Collapse of road from beach erosion and wave action (**left**) and wave overtopping during high tides (**right**) in the Seychelles. Source: B.G. Reguero.

*4.4. Other Effects of Climate Change in Coastal Areas*

In addition to flooding and erosion, sea-level rise will also produce other effects in coastal areas. The six main concerns for low-lying coasts from sea-level rise include (IPCC 2014): (i) permanent submergence of land by mean sea levels or mean high tides;

(ii) more frequent or intense extreme flooding; (iii) enhanced erosion; (iv) loss and change of ecosystems such as wetlands; (v) salinization of soils, ground and surface water; and (vi) impeded drainage.

Salinization is caused by rising sea levels that drive seawater intrusion into coastal aquifers, surface waters and soils. Salinization also increases with land-based drought events, decreasing river discharges in combination with water extraction and SLR. Seawater intrusion is already contributing to the conversion of land or freshwater ponds to brackish or saline aquaculture in low-lying coastal areas of southeast Asia, such as the Mekong Delta [65]. SLR will also affect agriculture mainly through land submergence, soil and fresh groundwater resource salinization and land loss from permanent inundation and erosion, with consequences on production, livelihood diversification and food security, especially in coastal agriculture-dependent countries, such as Bangladesh [66]. Salinization is already a major problem for traditional agriculture in deltas and low-lying island nations [67,68].

SLR may also affect tourism and recreation through impacts on landscapes (e.g., beaches), cultural features and critical transportation infrastructures, such as harbors and airports [69]. Coastal areas' future tourism and recreation attractiveness will, however, also depend on changes in air temperature, seasonality and sea surface temperature. Although ocean warming and acidification will be more influential in global fisheries and aquaculture, sea-level rise may produce indirect effects through adverse effects in habitats or facilities.

## 5. Responses to the Inevitable and Accelerating Rise in Sea Level and Coastal Hazards

With the thousands of kilometers of developed beaches, dunes, barrier islands, bluff and cliff tops around the planet, virtually every coastal nation has all types of developments, whether private or public, whether new or old, and land threatened by erosion and/or flooding in the decades ahead. What can or should be done with the communities and cities, homes and hotels, streets and parking lots, airports and power plants, wastewater treatment plants, pump stations and transmission lines or other infrastructure built on the beach or at the edge of a cliff or bluff? This challenge has affected and will increasingly affect nearly every coastal community on Earth and will only become more acute and costly over time [70].

There are a limited number of options, however, and all come with some costs, benefits and impacts. Depending on the location, some of these may require successfully navigating and negotiating through a complex, expensive and time-consuming permitting and environmental review process. Future losses will be high. The threat from future sea-level rise to coastal cities and low-lying areas around the world, combined with the storms, erosion and inundation, and the rapid degradation of natural coastal systems, will be one of the major societal and infrastructure challenges of this century [18,22,43,71–73]. Threatening levels of sea-level rise, however, are a longer-term issue, at least for now, but require mainstreaming adequate planning and rethinking of how coastal communities plan for new development in coastal areas as well as existing development, manage ecosystems and other coastal resources and prepare action to mitigate the impact of existing hazards, such as El Niño, hurricanes or tsunamis.

Throughout the 20th century, developed coastlines around the world have been responding to the hazards of shoreline flooding and coastal erosion or shoreline retreat in several ways:

- Do nothing (or wait and see);
- Beach nourishment or adding sand to beaches;
- Preventive actions in order to maintain the shoreline (i.e., hold the line) through either soft or hard solutions that may include armoring or hardening the shoreline;
- Managed or unmanaged retreat or realignment;
- Regulatory and restriction options on new development.

Each of these options has its positives and negatives and different geographic areas, political entities, communities, cities, states or nations have either intentionally or

unintentionally made decisions to use one or several approaches as short to intermediate term responses.

### 5.1. Do Nothing (or Wait and See)

This strategy may have the lowest cost upfront, but also the greatest risk of potential consequences. Whereas it is difficult to predict when any particular structure built on the back beach or on the edge of an eroding bluff will be inundated, damaged or collapse, doing nothing almost guarantees that the day will come when it's too late and damage or complete loss will result. This approach incurs no costs until a major event finally does occur, which usually cannot be predicted very far in advance, and then the losses may be high or catastrophic as last-minute protection might not be permissible, possible or effective. Depending on the setback of a particular structure from the shoreline or bluff edge, its elevation relative to sea level and wave runup (also known as freeboard), age or condition, past erosion or flooding problems, maintenance level and the future actions from local sea-level rise and storm impacts (which may exceed their original design conditions), this approach may work for a limited period of time.

### 5.2. Beach Nourishment

Approximately 80% to 90% of the sandy beaches along the U.S. Atlantic and Gulf coasts are experiencing erosion, with rates averaging 0.6 m per year (Heinz Center, 2000). While many factors contribute to shoreline recession, sea-level rise is the underlying factor accounting for the nearly ubiquitous coastal retreat [74]. This land loss has enormous economic impacts because some of the most expensive real estate in the United States is beachfront property.

One approach used for temporarily forestalling shoreline retreat or beach erosion is to artificially widen a beach with sand from some outside source, usually from the offshore continental shelf. Beach nourishment is usually carried out as very large-scale projects, where thousands of meters of shoreline are at least temporarily widened with thousands or millions of cubic meters of sand. Restoring beaches through beach nourishment can greatly increase their attractiveness to tourists [75]. Beach nourishment has been employed for decades along the low relief, typically barrier island-backed sandy shorelines of the Atlantic and Gulf coasts of the U.S. Over 1.35 billion $m^3$ of sand has nourished the beaches of 475 U.S. communities since 1923, at a 2020 real cost of USD 10.8 billion (https://beachnourishment.wcu.edu/, accessed on 1 August 2021). While several states have long-term beach management plans, the great majority of the funding for placing sand on beaches comes from the federal government through the Army Corps of Engineers.

Whether New Jersey, New York or Florida (USA), while literally billions of federal dollars have been spent moving sand from offshore to the shoreline for both recreational and shoreline protection benefits, the lifespan of the sand added artificially to these beaches in many cases has been relatively short, and in some instances has been less than a year. For some perspective on lifespans of individual nourishment projects, Florida (USA) has 15 beaches that have each been nourished 15 or more times, and Palm Beach has been nourished 51 different times (https://beachnourishment.wcu.edu/ accessed on 1 June 2021). It is clear that beach nourishment should not be seen as a permanent or even long-term solution to beach or bluff erosion, but simply buys a little more time at great public expense.

However, beach restoration can have important benefits too. The resurgence of Miami Beach, Florida, was largely attributed to saving the Art Deco architecture in South Beach, which was a significant achievement, but beach restoration was paramount to the recovery [76]. Beach nourishment in the late 1970s and early 1980s rejuvenated Miami Beach, which brought back the visitors and hence the economy [76,77]. The massive beach nourishment project, which was the largest such project of its kind undertaken in the world at that time, cost USD 51 million. White coral sand was pumped from deposits a few kilometers offshore at the cost of only a few dollars per cubic meter. Miami Beach was widened by approximately 60 m along the 16 km of barrier shoreline. This beach

nourishment project is often cited as the most successful beach restoration in the USA because of its longevity and positive economic impact. Most recently, Miami is investing USD 16 million in fresh sand to push back against erosion, while also maintaining a dune belt. The economic benefits of these projects are clear but face challenges from the costly maintenance, the duration and the rising threats from sea levels and storm action (Figure 10).

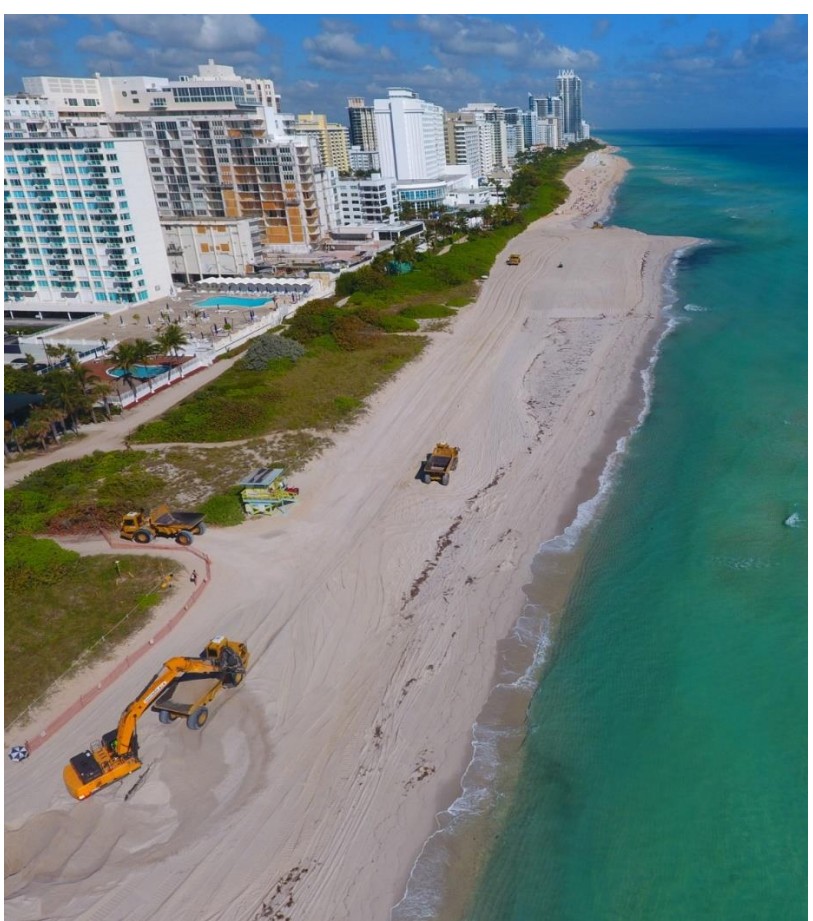

**Figure 10.** The U.S. Army Corps of Engineers dumps new sand from Central Florida (USA) along the Miami Beach shoreline. Source: U.S. Army Corps of Engineers.

In the USA, six states account for over 83% of the total volume of sand placed on beaches: California, Florida, New Jersey, North Carolina, New York and Louisiana [78]. The largest recipients have been Florida, New York and New Jersey, who have received approximately 500 million m$^3$ of sand since the 1930s, much of this funded by the federal government. Adding in the remaining Atlantic and Gulf Coast states, the amount of sand dredged from offshore and dumped on the beaches totals approximately 1.3 million m$^3$ (ASBPA). This is a difficult volume of sand to visualize but is enough sand to build a beach 50 m wide, 3 m deep and 8667 km long, or a beach extending all the way down the Atlantic seaboard from Maine around the southern tip of Florida and west across the entire state of Texas and beyond. Much of this sand has been moved around at federal expense and in recent years, with federal budgets being stressed, these projects have been more difficult to fund.

Overall, beach nourishment remains controversial. Shorefront cities and development have come to rely upon continuing projects funded by the federal government to restore and maintain beach values. While maintaining the beach is essential for recreation, the storm protection of property and beach-dependent economic activities, re-nourishment projects are a costly intervention, paid by taxpayer money and maintaining high real estate

interests for those who live by the beach, which can be considered a form of subsidy to the wealthy [79].

Traditional approaches to beach nourishment that merely add sand have also had important limitations in effectiveness. Beach replenishment success and environmental impacts may arise when one or more of the following factors is lacking: (1) a realistic assessment of potential borrow area sand volume; (2) compatibility of added sand to the beach being nourished, (3) construction costs; (4) the vulnerable geomorphic elements of the coastal zone; and (5) environmental impacts. Additionally, pre- and post-replenishment monitoring studies have frequently been inadequate in answering the questions of environmental impacts [80,81]. The ecological consequences of beach replenishment can also be significant [82]. In order to be effective, beach nourishment needs to be combined with sediment management techniques; ideally sand retention efforts [83,84], whether groins or some other mechanism to hold the sand in place so that it survives for a longer period of time and avoids frequent and costly sand feeding cycles. This requires understanding and characterizing the historic causes of erosion, either episodic or chronic, the long-term littoral drift rates and directions, and the natural processes [63,85].

With sea-level rise and increased wave action, beach nourishment will also need to be combined with other options, including adequate setback zones that can naturally nourish the system. Although the Army Corps of Engineers, as well as many local beach communities and coastal organizations, continue to put forward beach replenishment as a "soft" solution to coastal erosion or beach loss, other lines of evidence indicate that beach replenishment (alone) may not be a sustainable strategy in the long term to mitigate climate change [86]. For example, projections in California indicate that beach replenishment will only marginally delay the long-term inevitable loss of southern California beaches due to sea-level rise [38] (Figure 11).

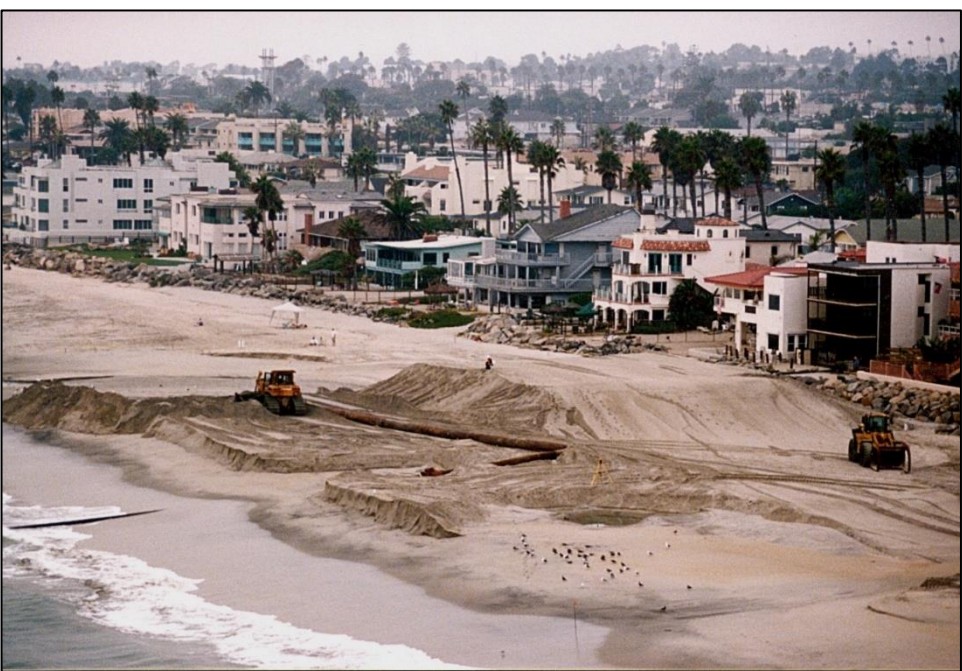

**Figure 11.** A number of beaches on the northern San Diego County coastline, California, have been nourished twice with a total of 2.7 million m$^3$ of sand at a total cost of USD 46 million. Beach surveys showed most of the sand had moved downcoast with littoral drift with a year or two (Courtesy: SANDAG Regional Beach Sand Project).

*5.3. Armoring or Hardening the Shoreline*

Whether rock revetments, seawalls, levees or floodwalls, or any of a variety of other engineered or non-engineered structures, hardening or armoring the shoreline has been

the most common historical approach to coastal erosion, shoreline retreat or flooding. These solutions aim to protect the shore by defending against elevated water levels or wave impacts. There is a long global history of coastal armoring, but in most cases, these structures were not built with their potential impacts to the surrounding environment and shoreline in mind. The potential effects of hardening the shoreline include visual impacts; loss of public beach due to placement of the structure on the beach; loss of the sand previously provided by the eroding cliff, bluff or dune being armored; passive erosion or the gradual loss of the beach fronting the armor with a continuing rise in sea level [39] (Figure 12).

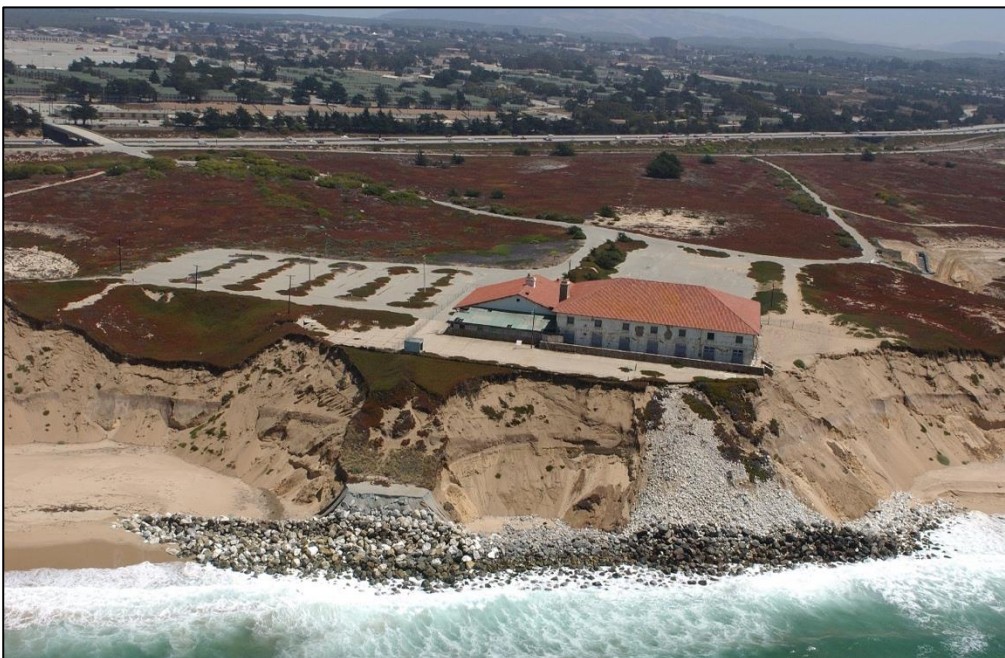

**Figure 12.** Passive erosion (loss of beach) in front of a rock revetment in central Monterey Bay, California, while a beach continues to exist on either side where there is no armor. Courtesy: California Coastal Records Project.

It is important to understand that coastal armoring (including seawalls and revetments) protects what is behind the armor, at the cost of the fronting beach. Combating erosion with a hard structure parallel to the shoreline is a choice to not protect the beach at that location. It is only a matter of time before beaches in front of hard armoring structures will be flooded or disappear with a rising ocean [39] Figure 12). Structures also produce shoreline encroachment and coastal habitat squeeze [24]. Coastal development and coastal armoring present physical barriers for the natural inland migration of coastal habitats, and changes in hydrological connectivity reduce sediment inputs and the potential for vertical accretion. Encroachment is also a term used to describe the advancement of structures, roads, buildings and other developments into natural areas, including the shoreline and the buffers around these areas. The term encroachment encompasses the placement of fill, the removal of vegetation or an alteration of the natural topography. This causes impacts to the functions and values of natural areas, such as water quality, loss of habitat (both aquatic and terrestrial), loss of flood attenuation potential or modifications in ecological processes.

However, there are many locations where coastal structures have historically been deemed necessary to protect critical infrastructure and other important assets. They will no doubt be used in the future to protect high value coastal infrastructure (large international airports, for example) that will be prioritized for protection for as long as possible. For many other oceanfront development locations, however, hard armoring structures will eventually become increasingly impractical, costly, unaffordable or unacceptable. The effectiveness

of existing armoring will vary depending upon the age, engineering, foundation depth, height and lateral extent of the individual seawall or revetment, as well as exposure to wave energy and elevated water levels. Overtopping of current armoring structures during moderate to extreme events may demonstrate the need for existing seawalls and rock revetments to be engineered to stand taller. As proposed seawalls become larger and stronger, this will inevitably bring new concerns and conflict in the regulatory process concerning increased environmental impacts.

In the USA, some coastal states have essentially banned any new hard structures altogether, while others have made it more and more difficult to obtain a permit unless the primary structure (a house for example) is under imminent threat. The era of routine armoring of any eroding stretch of coastline in the United States is ending, as the negative impacts of protective structures have been increasingly documented, recognized and understood and the inevitability of future sea-level rise has become more obvious [40,84]. While armor can provide short- or intermediate-term protection for private property and public infrastructure, with a changing climate and a rising sea, there are no future guarantees that today's armor will survive far into the future. A review of research and experience also demonstrate that there exist a range of financial, policy, planning and management tools, often used for different purposes, that can be readily implemented or modified to address coastal squeeze and enable inland habitat migration. Awareness of approaches/solutions can assist in accommodating the migration of habitats as a necessary component of coastal management in an era of increasing rates of sea-level rise.

### 5.4. Soft Protection Approaches and Working with Natural Processes Rather Than against Them

The rising costs of coastal armoring and its intrinsic challenges in a changing climate has driven an increasing interest in other soft approaches. These approaches include a growing number of 'engineering' and 'building with nature' initiatives that leverage natural ecosystems for their coastal protection value [87–90]. Examples of "soft" approaches are increasing and include managed sediment relocation and beach and dune restoration with plants and landscape reshaping, as well as other living shorelines alternatives (e.g., oyster reefs). For example, the Spanjaards Duin in the Netherlands is one of the first examples of constructing artificial dunes in order to create natural dune habitats as a compensation measure for port development. This project leverages natural processes to shape the dunes by ensuring proper grain size for aeolian dynamics and maintaining the groundwater level to support vegetation. Experiences on the U.S. Gulf Coast also demonstrate successes with soft protection in low energy environments [90]. Although the number of examples is growing rapidly, the experience base and dynamic nature of these solutions makes monitoring a critical component in maintaining and expanding soft coastal protection solutions.

However, along many coastlines where coastal ecosystems provide protection, one important strategy is maintaining and enhancing this "natural infrastructure". The term "natural infrastructure" recognizes the value that ecosystems, such as coral reefs, mangroves, salt marshes, beach and dune systems and vegetation (emerged or submerged), offer by attenuating waves, reducing the impacts of sea-level rise and retaining sediment or building land [91,92]. For example, reefs [22,70,93–95], seagrasses [96] or saltmarshes and beach and dune systems can be effective in protecting against the impact of storms [63,97,98] under certain conditions. Such protection can be very valuable. Recent valuations demonstrate that U.S. reefs provide over USD 1.8 billion in flood mitigation benefits every year, with much of the protection being highly concentrated in certain reef-lined coastlines, including the most vulnerable communities in island territories [22]. For many of these coastlines, including tropical nations and small island states at the forefront of the impacts of climate change, maintaining this natural infrastructure may be one of the most cost-effective adaptation strategies, at least over the short term.

Many organizations in civil works research, development and technology programs, as well as conservation and environmental organizations, are increasing the momentum

and knowledge base for embracing nature-based coastal adaptation to more sustainably deliver economic, social and environmental benefits [99–101]. However, some of these ecosystems are restricted to particular latitudes and energy conditions (Figure 13). For example, salt marshes and similar wetland ecosystems require fine-grained sediment and cannot tolerate or survive high wave energy conditions. Where wave energy is low, such as estuary and lagoon environments, vegetation can be protected and even restored. However, living shorelines, such as vegetated shorelines and dunes, also have limitations and can be eroded or removed under severe wave and storm conditions (Figure 14). In the long term, these approaches will also be affected by long-term sea-level rise, unless they have the space and conditions to self-adapt (e.g., hydrological conditions, sediment supply, etc.). To provide context, throughout the millions of years that these coastal ecosystems have existed, they have been able to keep pace with sea level fluctuations of more than 100 m, as evidenced by the very existence of these habitats or ecosystems today. However, added pressures on these ecosystems currently threatens their potential to keep pace with sea-level rise and affects their resilience to storm impacts [17,102–104], therefore threatening their future effectiveness for flood and erosion reduction. For example, where the landward edges of these coastal environments are urbanized or developed (i.e., coastal squeeze), eventual shoreline migration with future sea-level rise will be compromised.

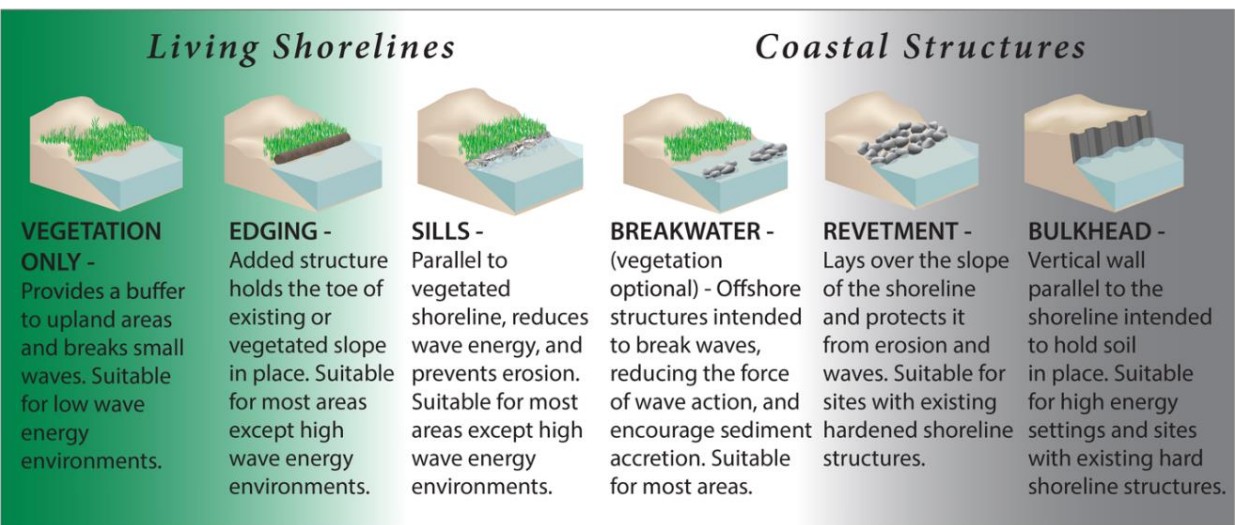

**Figure 13.** Range in coastal protection approaches from green to gray with those on the left ("living or partially living") suitable only for low wave energy environments. (Source: NOAA, [88]).

### 5.5. Managed Retreat, Realignment and Setbacks

Climate-induced relocation and managed retreat are increasingly considered as part of the adaptation planning process in many coastal areas, although deciding whether, where, when and how to move is very complex and controversial. Managed retreat or managed realignment is a coastal management strategy that involves the controlled flooding of low-lying coastal areas and the abandonment or relocation of assets and people allowing the shoreline to move inland, instead of attempting to "hold the line" [98]. Managed retreat may include various approaches: planning and setback zones; relocation of buildings; buy-back and buy-out programs; buy-out and rental programs; setbacks (horizontal or vertical) with regulations on new development or no-build areas; restoration of shoreline and habitats; green spaces and controlled flooding areas and erosion setbacks. For example,

flood defenses can be set back from the shoreline in order to allow erosion or flooding in certain areas. Often, this approach also offers opportunities for enhancing coastal habitats and their ecosystem services, including coastal protection.

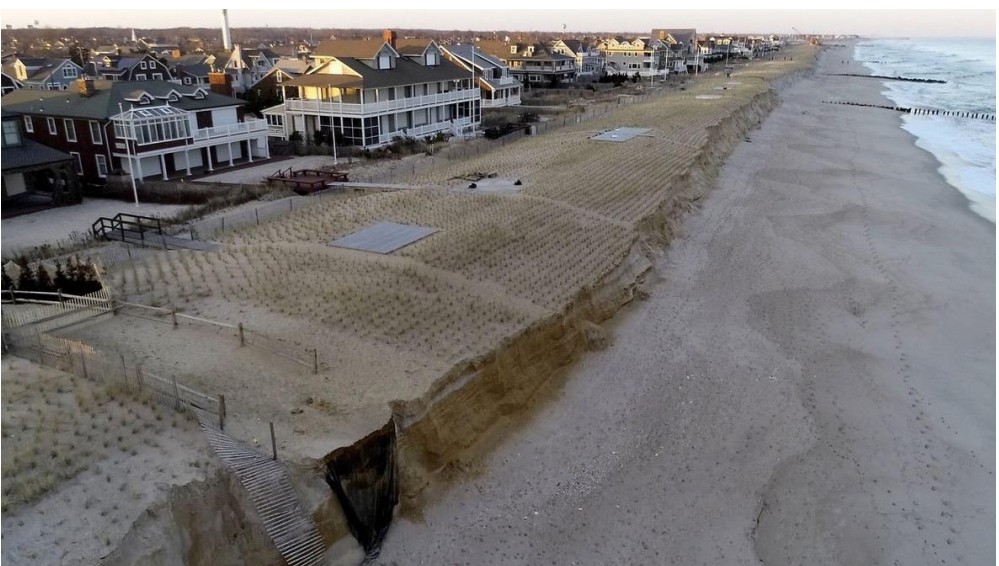

**Figure 14.** Storm waves along the U.S. Atlantic coast have removed a major portion of a newly built and vegetated dune at Bay Head, New Jersey (Photo: Mike Davis and Thomas P. Costello).

Many cities are beginning to discuss, consider and plan for how they will adapt to higher sea levels, while others are resisting. For example, the state of California in the USA has invested significantly in planning to respond to sea-level rise, and managed retreat is being actively debated in dozens of communities [105]. Although managed realignment and sacrifice zones are also a form of battling flood risk and erosion [106], managed retreat requires interweaving science and governance for community decision-making [107]. Some important factors affecting the feasibility of managed retreat include [108,109]: presence of coastal defenses, availability of low-lying land, flood or coastal defense need, coastal management frameworks, intertidal habitat benefits and societal awareness and acceptance.

## 6. Deciding the Strategy and Planning Long-Term Responses

Broadly speaking, the main responses in coastal adaptation are to protect, manage/accommodate, and retreat [1]. Retreat can be pre-emptive, just-in-time or reactionary. Pre-emptive planned retreat or proactive abandonment would involve the systematic relocation of buildings or communities well before they are impacted by major coastal flooding or threatened by cliff or bluff erosion. Just-in-time retreat involves delaying retreat as long as possible, but prior to major damage or loss. This approach would take place when many of the involved property owners or entities realize the risk is unacceptable or when the sea level reaches some threshold level that has been established and agreed upon in advance. Reactive retreat can be implemented following a major flooding or disaster event, and could be designated as unplanned retreat, where this is the only option left. This would require local, state or national government legislation preventing reconstruction in high-risk areas that have recently been devastated and potentially implementing some type of partial buyout program.

Along many coastlines it is only a matter of time before resisting the advance of the sea will no longer be possible. Whether through soft solutions, such as living shorelines, beach replenishment or hard armoring, maintaining the present position of the shoreline will become infeasible in many areas. Timing may depend on the local hazards, consequences, existing protection, management plans, funding and geomorphology of each coastline, but

it is not a question of whether it will happen, but when. In many communities, it is already happening [45,46,110].

There is an alternative and longer-term approach. With sea-level expected to rise between ~50 cm (~20 inches) and ~310 cm (~10 feet) by 2100, and continuing on for centuries into the future, and also with changes in wave action and hurricane frequency and magnitude expected, virtually every shoreline development, community and city in the world needs to begin to think for the long-term. Communities can start by looking at current management practices, the history of storm damage and erosion, the exposure or vulnerability of both public infrastructure and private development. and both present and future hazards and challenges [111]. Where structures are located in vulnerable coastal locations and have experienced or are threatened by frequent flooding or cliff retreat, relocating or removing the structure is going to become a more realistic and important consideration or response. There are many existing developments where there are no other reasonable or acceptable alternatives, where future damage or destruction is almost guaranteed and where rebuilding or protection is simply not possible or cost-effective. In these coastlines, managed retreat should be considered, as well as the associated costs and opportunities (see previous sections). Certainly, the size, condition and physical setting of the building or infrastructure will be important considerations. However, there are still a limited number of examples of successful relocation or planned retreat, whether homes or other structures. The Cape Hatteras Lighthouse in North Carolina (USA) is a good example of what can be done for USD17.5 million: the tallest lighthouse in the USA, weighing 4360 tonnes, was moved inland 887 m (2900 feet) in 1999. "Hold the line" coastal protection, in many other instances, can buy time to prepare other responses to realign or relocate critical infrastructure (e.g., airports, power plants, wastewater treatment facilities, etc.).

The choice between protect, manage/accommodate and retreat will be a local one and undoubtedly include a range of options. Coastal regulation so far has largely failed in part by not including projected rates of sea-level rise and erosion, and could be improved by consistent political oversight, will and enforcement [112]. New initiatives, such as multiple response pathways, can provide flexibility in difficult adaptation investment choices, which can be more adequate from a socioeconomic perspective than a single adaptation strategy. Adaptation pathways are defined as a sequence of adaptation actions or strategies over time, such as beach nourishment, building new levees and floodproofing or elevating buildings, which anticipate uncertain and changing risk conditions, such as sea-level rise [113,114]. Identifying "investment tipping points", after which a transition could decrease the economic efficiencies, should be a critical part of adaptation planning [115]. How much risk is tolerable and when a pathway should be taken is also a complex issue. For example, there is also the increasing issue of obtaining insurance for at-risk coastal homes along the U.S. Atlantic coast, where insurers are beginning to cancel or no longer insure high-risk properties. Without insurance coverage, however, homeowners will not be able to obtain mortgages, which is beginning to further impact the risks facing coastal homeowners. Economics is beginning to lower home values along the Eastern Seaboard. The First Street Foundation (2019) conducted housing market research for 18 states along the East and Gulf Coasts, from Maine to Texas.

Data show that increased tidal flooding driven by sea-level rise lowered property values by USD 15.9 billion between 2005 and 2017. Among the states evaluated, Florida has witnessed the greatest loss in relative home value at USD 5.4 billion, followed by New Jersey at USD 4.5 billion, New York with USD 1.3 billion and South Carolina with USD 1.1 billion. As property values have continued to decline and extreme events and tidal flooding have gradually flooded and damaged more cities more frequently, insurance companies are now looking carefully at which policies to cancel, which properties not to insure, and for those that appear to be insurable, what are realistic premiums that will cover projected losses. These forces are all well beyond the homeowners' power to influence and will have increasingly negative effects on both insurance and loans for coastal homeowners.

### 7. Challenges to Adaptation

Local, regional, central and federal governments need to continue their efforts to reduce greenhouse gas emissions, but it is also important to adequately prepare for the climate impacts that are already underway. Mitigation is needed to reduce and slow down global warming and sea-level rise, but adaptation has to be part of the solution to climate change, as there are inevitable impacts that societies will have to face, especially along coastlines where so many natural and socioeconomic interests intersect. However, adaptation to sea-level rise has found key types of constraints [23,116] that involve technological, social conflict, economic and financial barriers and limitations.

One of the critical challenges for upscaling adaptation efforts is finance. Many governments, especially in developing countries, need to balance pressing needs with limited budgets, especially during the global pandemic recovery. A revision of climate finance in 2019 determined that climate finance flows in the 2017–2018 cycle left USD 30 billion investments in adaptation versus USD 537 billion in mitigation [117]. Most of the adaptation investments fell within programs dedicated to disaster risk management, water and waste management, and land use. This also shows the cross-sectoral nature of adaptation. However, adaptation financing rose significantly from its previous level in 2015/2016, making up only 5.6% of funding and with no change from 2015/2016 as a percentage of tracked finance. Multilateral funding is the second source of adaptation funding after national sources. Globally, the multilateral development banks collectively committed USD 61,562 million in climate finance in 2019—USD 46,625 million (76% of total for climate change finance) and USD 14,937 million (24% of total) for climate change adaptation finance [118]. Therefore, while climate finance is steadily increasing, it still falls far short of the needed investments and makes clear the large global adaptation funding gap.

Economic barriers are another critical limiting factor. They reflect the complex balance between socioeconomic benefits and costs of adaptation versus the impacts they avoid. Cost benefit analyses, which are prescribed for coastal projects in some countries, such as the U.S., can be challenged by limitations related to the difficulty or impossibility to monetize benefits, the consideration of both long time-frames and materialization of the benefits and the potential differences arising from public versus private interests and social versus individual preferences, including the social discount rates considered.

Technological limits refer to when there are no adaptation options available to effectively reduce the impacts of SLR, or when they are not enough to keep risk at acceptable levels [119]. Protection and accommodation measures have been in societies for centuries, as communities have used coastal engineering to adapt to environmental change and local hazards, e.g., seawalls from the colonial era in the Indo-Pacific. Furthermore, some technologies are not feasible or possible in some countries and coastal environments. There are also limitations in the local availability of materials, knowledge or construction, especially in developing countries and small island states. Another important technological aspect is the capacity and ability to maintain, repair and rebuild adaptation infrastructure beyond its construction. In many developing regions, for example, adaptation projects are implemented by external parties and leave no local capacity or knowhow to maintain or expand such measures over time.

Coastal adaptation is also a clear collective action problem: a group benefits from an intervention but no individual has the sufficient incentive to act alone [120]. Social conflict barriers may arise whenever stakeholders hold conflicting interests that may be overcome through governance to develop norms, laws or policies in order to resolve conflicts or achieve gains [121]. Formal institutions can help, but other informal means and community involvement is needed. Social barriers remain challenging and difficult to overcome and can be further aggravated when local needs, contexts and priorities are not adequately considered [122]. For example, solutions that are considered effective in some regions can be ignored or perceived negatively in other areas. One clear example is found in low crested and submerged structures that have been amply used in Europe, including with environmental benefits [123], but can be perceived as negative in, for example, tropical

regions. Social challenges to adaptation can also be driven by diverging interests between parties, including who benefits and who pays, which can be a major decision factor when planning adaptation. Other social barriers are related to conflicts around cultural and social priorities and constraints.

Barriers to adaptation can be overcome through efforts to address gaps in technology, economic and human resources, management and institutional change. Understanding risk, planning and finance adaptation are key aspects for accelerating adaptation [124]. Furthermore, the process known as "Paris alignment", which refers to the alignment of finance flows consistent with a pathway towards low greenhouse gas emissions and climate-resilient development (article 2.1c, Paris Agreement), also represents an opportunity to expand adaptation efforts across activities in the global coastal zones.

## 8. Conclusions

Given the inadequacies to date in our efforts to significantly reduce global greenhouse gas emissions, and the reality that anything we are able to implement or accomplish in the near future to mitigate climate change and its impacts will not have immediate effects, we must also continue to direct our energy and creative thinking towards adaptation strategies that can prepare the coastlines and cities of the world for sea-level rise.

There is no question that sea levels are rising globally, and the uncertainties revolve around not "if" it is rising, but rather how much it might rise in specific locations and at what future times. The science is clear; the future is, unfortunately, increasingly certain too; the need to act now is urgent. There are no simple answers or solutions, but we need to involve all stakeholders in planning and implementing well-thought-out plans and policies for the inevitable future, with agreed-upon thresholds for when action will be taken.

**Author Contributions:** G.G. and B.G.R. jointly developed the conceptualization, review and writing of this article. Both authors have read and agreed to the published version of the manuscript.

**Funding:** This research received no external funding. B.G.R acknowledges the support from an Early-Career Research Fellowship from the Gulf Research Program of the National Academies of Sciences, Engineering, and Medicine, although the content is solely responsibility of the authors and does not necessarily represent the official views of the Gulf Research Program of the National Academy of Sciences, Engineering and Medicine.

**Institutional Review Board Statement:** Not applicable.

**Informed Consent Statement:** Not applicable.

**Data Availability Statement:** Not applicable.

**Conflicts of Interest:** The authors declare no conflict of interest.

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
