# Peer review of "Coastal Adaptation to Climate Change and Sea-Level Rise"

_water, doi:10.3390/w13162151_

Round 1

Reviewer 1 Report

The manuscript water-1293533, entitled “Coastal Adaptation to Climate Change and Sea-Level Rise”, aimed to review the effect of sea level rise and the status on adaptation to climate change in coastal zones. Such topic is interesting and meaningful, as it is helpful for people understanding the negative effects of climate change on coastal areas and the main challenges to adaptation, then deciding the strategy and planning Long-Term Responses.

The manuscript appears to be well-organized, however, some errors also exist. The current review is not suitable for publication in terms of presentation, content, and description in manuscript.

In introduction, the authors mainly described the climate change, the sea level rise phenomenon, and the methods of measuring sea level, however, the objective of the manuscript cannot find in this part. The aim of the manuscript is indispensable, because it is the thread of the whole article.

Besides, the manuscript principally show the effects of climate change and sea level rise in coastal areas, however, the affected coastal areas that the authors took examples in this manuscript are principally in U.S.. As we know, there are many coastal areas in the world, maybe the authors should not limit the scope of coastal areas in US.

Finally, the review at its current stage is lacking of insightful information. The authors just briefly described what happened in the past or what has been done by previous literature studies. Further comments and discussion are encouraged.

Some specific comments about this topic as follows.

  1. In the manuscript, before the section 7, we cannot find the section 5 and section 6. Where's section 5 and section 6?
  2. The provenance of the photograph (such as Figure 5, Figure 6, Figure 7, Figure 9, Figure 14) should be identified.
  3. In order to better distinguish, authors are advised to use a distinctly different color line in Figure 1.
  4. Line 168-171, the sentence should be revised.
  5. Line 159. there was only a “)”behind the “Project”, was it wrong? Please check it.

Author Response

Responses to Comments from Reviewer No. 1

The manuscript water-1293533, entitled “Coastal Adaptation to Climate Change and Sea-Level Rise”, aimed to review the effect of sea level rise and the status on adaptation to climate change in coastal zones. Such topic is interesting and meaningful, as it is helpful for people understanding the negative effects of climate change on coastal areas and the main challenges to adaptation, then deciding the strategy and planning Long-Term Responses.

The manuscript appears to be well-organized, however, some errors also exist. The current review is not suitable for publication in terms of presentation, content, and description in manuscript.

In introduction, the authors mainly described the climate change, the sea level rise phenomenon, and the methods of measuring sea level, however, the objective of the manuscript cannot find in this part. The aim of the manuscript is indispensable, because it is the thread of the whole article.

Response:

A short paragraph has been added at the beginning of the Introduction to explain the objective of the manuscript, essentially to provide a comprehensive overview of this Special Issue. We would like to note that this paper is not another article for WATER but a summary of the topic to precede the included papers in the special issue.

Besides, the manuscript principally shows the effects of climate change and sea level rise in coastal areas, however, the affected coastal areas that the authors took examples in this manuscript are principally in U.S. As we know, there are many coastal areas in the world, maybe the authors should not limit the scope of coastal areas in US.

Response:

The scope is not limited to the U.S. We give examples from the U.S., Africa, Asia and tropical small island states. This has been indicated in the revised introduction.

Finally, the review at its current stage is lacking of insightful information. The authors just briefly described what happened in the past or what has been done by previous literature studies. Further comments and discussion are encouraged.

Response:

We don’t understand this comment or what is lacking in the way of “insightful information” or that “further comments and discussion are encouraged”. Note that the aim of this article is to provide a summary review of coastal adaptation, to precede the rest of the papers in the special issue.

Some specific comments about this topic as follows.

  1. In the manuscript, before the section 7, we cannot find the section 5 and section 6. Where's section 5 and section 6?

Response: The heading numbers have all been corrected.

  1. The provenance of the photograph (such as Figure 5, Figure 6, Figure 7, Figure 9,Figure 14) should be identified.

Response: Credits for all of the photographs have now been included.

  1. In order to better distinguish, authors are advised to use a distinctly different color line in Figure 1.

Response: This is not possible because this was taken from a previously published National Academy of Sciences Report. The reference and credits are included.

  1. Line 168-171. The sentence should be revised.

Response: Sentence has been revised for clarity.

  1. Line 159. There was only a “” behind the “Project”, was it wrong. Please check it.

Response: We cannot find this orphan on line 159.

Reviewer 2 Report

It is a nice review paper. I’ve no comments related with the content. From my point of view is excellent. May be it would be nice to explore the backwards displacement policy in a parallel way to the solutions that consist just on preventing erosion, despite the green or the grey epithet. But it could be another paper…

I recommend accept the paper but before I suggest some minor edition revision:

  • Please the text is plenty of non-SI units. Since you write for an international audience, please adopt the SI and avoid, inches, feets, miles per hour, etc. There are many cases all along the MS.

  • Please the audience is international, so the care because you cite many USA cities without indicating the geographical location. You should use New York (USA), because when you comment foreign case, you indicate the State/Country. For instance when you deal with the Senegal cases.

  • Related to the SI units and cultural uses, there are a couple of cases where you use billion. We must be sure if they are American billion (10^9) or British one (10^12). Please clarify.

  • Check references, there are some of there that do not follow the reference style

  • I’m not a native English speaker, but there are few typing errors (i.e. othe instead other) .

I enclose a pdf with yellow marks on units, typing and references related to the former comments.

Author Response

Responses to Comments from Reviewer No. 2

It is a nice review paper. I’ve no comments related with the content. From my point of view is excellent. Maybe it would be nice to explore the backwards displacement policy in a parallel way to the solutions that consist just on preventing erosion, despite the green or the grey epithet. But it could be another paper…

Response:

We agree although in lines 594-685 we include a discussion of Managed Retreat and some of the obstacles or challenges to this adaptation approach. This is a topic with much being written at present so we wanted to include it but intentionally kept it brief as this summary paper is already 25 pages long

I recommend accept the paper but before I suggest some minor edition revision:

Please the text is plenty of non-SI units. Since you write for an international audience, please adopt the SI and avoid, inches, feets, miles per hour, etc. There are many cases all along the MS.

Response:

We have gone carefully through the entire paper and made sure that all units are now in SI units.

Please the audience is international, so the care because you cite many USA cities without indicating the geographical location. You should use New York (USA), because when you comment foreign case, you indicate the State/Country. For instance when you deal with the Senegal cases.

Response:

Geographic locations, for example: California (USA) have now been added for specific USA places, such as states.

Related to the SI units and cultural uses, there are a couple of cases where you use billion. We must be sure if they are American billion (10^9) or British one (10^12). Please clarify.

Response:

We have clarified the use of billion as American (109)

Check references, there are some of there that do not follow the reference style

Response:

All references have now been placed in the standard style for WATER.

I’m not a native English speaker, but there are few typing errors (i.e. othe instead other) .

Response:

Spell check was run again to correct any spelling errors.

I enclose a pdf with yellow marks on units, typing and references related to the former comments.

Response:

The marks in the pdf have been attended.

Round 2

Reviewer 1 Report

All the comments have been addressed.